# Changes in mGlu5 Receptor Signaling Are Associated with Associative Learning and Memory Extinction in Mice

**DOI:** 10.3390/life12030463

**Published:** 2022-03-21

**Authors:** Ana Elena Teleuca, Giovanni Sebastiano Alemà, Paola Casolini, Ilaria Barberis, Francesco Ciabattoni, Rosamaria Orlando, Luisa Di Menna, Luisa Iacovelli, Maria Rosaria Scioli, Ferdinando Nicoletti, Anna Rita Zuena

**Affiliations:** 1Department of Physiology and Pharmacology, Sapienza University of Rome, 00185 Rome, Italy; anaelena.teleuca@uniroma1.it (A.E.T.); sebastiano.alema@uniroma1.it (G.S.A.); paola.casolini@uniroma1.it (P.C.); barberis.1741985@studenti.uniroma1.it (I.B.); francesco.ciabattoni@uniroma1.it (F.C.); rosamaria.orlando@uniroma1.it (R.O.); luisa.iacovelli@uniroma1.it (L.I.); ferdinando.nicoletti@uniroma1.it (F.N.); 2IRCCS Neuromed, 86077 Pozzilli, Italy; luisa.dimenna@neuromed.it (L.D.M.); stabulario@neuromed.it (M.R.S.)

**Keywords:** mGlu5 receptors, polyphosphoinositide hydrolysis, VU0360172, spatial learning, extinction, hippocampus, prefrontal cortex, Gαq, PLC-β, norbin

## Abstract

Using an in vivo method for the assessment of polyphosphoinositide (PI) hydrolysis, we examine whether spatial learning and memory extinction cause changes in mGlu5 metabotropic glutamate receptor signaling in the hippocampus and prefrontal cortex. We use the following five groups of mice: (i) naive mice; (ii) control mice exposed to the same environment as learner mice; (iii) leaner mice, trained for four days in a water maze; (iv) mice in which memory extinction was induced by six trials without the platform; (v) mice that spontaneously lost memory. The mGlu5 receptor-mediated PI hydrolysis was significantly reduced in the dorsal hippocampus of learner mice as compared to naive and control mice. The mGlu5 receptor signaling was also reduced in the ventral hippocampus and prefrontal cortex of learner mice, but only with respect to naive mice. Memory extinction was associated with a large up-regulation of mGlu5 receptor-mediated PI hydrolysis in the three brain regions and with increases in mGlu5 receptor and phospholipase-Cβ protein levels in the ventral and dorsal hippocampus, respectively. These findings support a role for mGlu5 receptors in mechanisms underlying spatial learning and suggest that mGlu5 receptors are candidate drug targets for disorders in which cognitive functions are impaired or aversive memories are inappropriately retained.

## 1. Introduction

The type-5 metabotropic glutamate receptors (mGlu5 receptors) play a crucial role in activity-dependent and homeostatic mechanisms of synaptic plasticity, such as long-term potentiation (LTP), long-term depression (LTD), metaplasticity, and synaptic scaling [1,2,3,4]. MGlu5 receptors have been extensively studied in the hippocampus, where LTP and LTD reflect cellular modifications that occur during formation/storage and the extinction of memories, respectively [5]. The hippocampus does not act as a unitary structure, with the dorsal and ventral portions displaying different functions [6,7]. The dorsal hippocampus is the “cognitive” portion of the hippocampus, and its lesion in rodents severely impairs spatial memory [8]. In contrast, the ventral hippocampus is involved in the responses to stress and emotional behavior [9], and its lesion does not prevent spatial memory in the water maze test [8].

The mGlu5 receptors mediate neuronal oscillations that allow information transfer within the hippocampus [10,11] and mediate hippocampus-dependent spatial memory [12,13,14]. Accordingly, mGlu5 receptor knockout mice show impaired learning and memory in the Morris water maze [15], and similar findings were obtained following specific deletion of mGlu5 receptors in the dorsal hippocampus [16]. In addition, the systemic administration of two selective positive allosteric modulators (PAMs) of mGlu5 receptors enhanced memory acquisition and consolidation in the Morris water maze [17,18], whereas the impaired performance in spatial learning and working memory tests was observed following pharmacological blockade of the mGlu5 receptor [10,19].

The involvement of mGlu5 receptors in memory-related processes may not be restricted to memory acquisition and consolidation, but may be extended to a process known as extinction learning, a form of inhibitory learning that contributes to behavioral adaptability and flexibility. Extinction represents the ability of an individual to learn that a previously acquired association is no longer valid. This might be considered a process of new learning that suppresses the original association formed during acquisition. Several studies have shown that potentiating and inhibiting mGlu5 receptor function results in facilitation and suppression of extinction, respectively. For example, mice lacking mGlu5 receptors show impaired extinction in the Morris water maze [20], and mGlu5 receptor blockade with MPEP prevents extinction of both fear conditioning and spatial memory [21,22]. In contrast, pharmacological activation of mGlu5 receptors with the PAM, ADX47273, enhances extinction [23]. These findings have a potential translational value because extinction appears to be disrupted in various psychiatric disorders, including attention-deficit hyperactivity disorder (ADHD) [24], anxiety [25], schizophrenia [26], and post-traumatic stress disorders (PTSD) [27]. 

What is only partially known is how mGlu5 receptor expression changes in the hippocampus during learning and in response to extinction. In rats subjected to a fear-conditioning paradigm, we found an early increase in mGlu5 receptor protein levels in the hippocampal CA3 region and a later increase in the CA1 region and dentate gyrus after acquisition training; overexpression of mGlu5 receptors in CA1 was also observed after several days of extinction [28]. In addition, an increased expression of mGlu5 receptors was found in the hippocampus at 24 or 48 h following the induction of LTP at the perforant pathway-dentate gyrus synapses [29]. To our knowledge, there are no data on mGlu5 receptor signaling in response to learning and extinction. The mGlu5 receptors are coupled to Gq, and their activation leads to phospholipase-Cβ-mediated polyphosphoinositide (PI) hydrolysis, with the ensuing formation of inositol-1,4,5-trisphosphate (InsP_3_) and diacylglycerol [30]. Most of the studies of mGlu5 receptor-mediated PI hydrolysis have been performed using brain slices preloaded with radioactive inositol and then treated with micromolar concentrations of lithium ions to block the conversion of inositol monophosphate (InsP), the final metabolite of InsP_3_, into inositol. This method has a number of limitations, including the need to pool tissue from several animals, the need to perform ex vivo experiments in slices, and the lack of information on endogenous InsP levels (and, therefore, on the specific activity of ^3^H-InsP). We have developed a new method for the in vivo assessment of mGlu5 receptor-mediated PI hydrolysis, based on systemic treatment with LiCl followed by the mGlu5 receptor PAM VU0360172 and determination of endogenous InsP levels by ELISA [31]. With the aid of this method, we have decided to study how mGlu5 receptor signaling is affected by spatial learning and by extinction in the dorsal and ventral hippocampus and in the prefrontal cortex. We have also measured protein levels of mGlu5 receptors, the α subunit of Gq protein, phospholipase-Cβ, and the mGlu5 receptor-interacting protein, norbin [32,33]. 

## 2. Materials and Methods

### 2.1. Animals

The experiments were carried out on adult CD1 male mice (7–8 weeks of age) (Charles River, Italy). All mice were housed in a controlled-temperature room (21–23 °C, humidity 40–50%) and maintained on a 12 h light/dark cycle with food and water *ad libitum*. All efforts were made to minimize the number of animals and to alleviate their discomfort. All experimental procedures were performed in conformity with the European Union Directive (2010/63/EU) on the protection of animals used for scientific purposes and were approved by the Italian Ministry of Health (DDL 26/2014 and previous legislation; protocol number n. 1030/2020-PR).

### 2.2. Experimental Design 

We performed 4 different experiments. In experiments 1 and 2 we tested the following three groups of mice: naive, controls, and learners; in experiment 3 we tested the following five groups: naive, controls, learners, “extinction,” and “forgetting”; in experiment 4 we tested only the following two groups: extinction and forgetting. Naive mice were left undisturbed in their cages. Control mice swam for an equivalent amount of time as compared to the trained groups, but without learning (the escape platform was absent); this group was included to ensure that changes in mGlu5 signaling would be specifically learning-related. Learner mice were subjected to learning the position of a submerged platform (acquisition and retention). Extinction mice were subjected to learning the position of the platform (acquisition and retention) and subsequently, to a forced extinction of the acquired memory. Forgetting mice were subjected to the acquisition and retention phases of spatial memory and were subsequently left undisturbed in their cages for 5 or 7 weeks in order to allow a spontaneous extinction of the learned task. See Figure 1 for an experimental diagram of the study.

### 2.3. Morris Water Maze

The apparatus consisted of a circular black pool (diameter 120 cm, height 60 cm) located in a test room with white walls and many cues on them. The pool was filled to a depth of 40 cm with water (kept at 21 ± 1 °C), covering an invisible, black, 10 cm square platform. The platform was about 1 cm beneath the water’s surface. The pool was virtually divided into four quadrants with the platform positioned in the center of a quadrant. The mice were placed on the platform for 30 s on the first day of testing (before starting the test) so that they could detect external cues and orient themselves. 

Spatial training (acquisition). Mice were subjected to five trial training sessions, spaced by 24 h, for four consecutive days (days 1–4). In each trial, mice were permitted to swim until they reached the escape platform and climbed on it. A 60 s cut-off was chosen, after which the mice were placed on the platform and left for a reinforcement period of 30 s. The starting location of the animal was different in each trial and the platform quadrant was avoided. The amount of time each animal spent to reach the platform (escape latency) was measured. 

Probe (memory retention). The retention of the spatial training was assessed on day 5 (24 h after the last training session) through a single probe trial in which the platform was removed and the mice were left to swim for 60 s; the time spent in the quadrant where the platform was previously positioned (target quadrant) was evaluated. 

Extinction. Forced extinction of the acquired memory was induced two days after the probe trial (two days in which the animals were left undisturbed), by exposing mice to non-reinforced swimming tests, therefore in the absence of platform. Mice were subjected to two 60 s trials for 3 consecutive days (days 6–8) (with a 30 min inter-trial interval) and the time spent in the quadrant where the platform was previously positioned (target quadrant) was measured.

Forgetting. Spontaneous forgetting was assessed on day 37 or 54 (5 and 7 weeks after the acquisition phase, respectively) in which mice underwent a single 60 s probe trial (in absence of the platform). The time spent in the target quadrant was measured. 

Behavioral data from the training, probe, extinction, and forgetting sessions were acquired and analyzed using an automated video-tracking system (ANY-Maze, Stoelting, Wood Dale, IL, USA).

### 2.4. Drugs and Treatment 

Thirty minutes after the last behavioral session, all animals were pretreated with lithium chloride (LiCl) (100 mg/kg i.p.) (Sigma-Aldrich, Milan, Italy), a dose that was expected to increase receptor agonist-stimulated InsP formation due to lithium’s ability to inhibit the conversion of InsP into free inositol [34]. Mice were given PAM VU0360172 (30 mg/kg) or vehicle one hour after being treated with lithium chloride. One hour after drug (VU0360172 or vehicle) injections, mice were killed by decapitation, and the prefrontal cortex, dorsal, and ventral hippocampus were quickly dissected and stored at −80 °C until InsP measurements and western blot analysis (See Figure 1 for the experimental diagram and Figure 2 for a schematic representation of prefrontal cortex dissection). Lithium chloride was dissolved in saline and VU0360172 (Tocris Bioscience, United Kingdom) was dissolved in 10% Tween 80 and adjusted to pH 7.4 with NaOH. Drugs and vehicles were administered i.p. in a volume of 5 mL/kg body weight. The dose used for VU0360172 (30 mg/kg) was selected on the basis of our previous data [31]. 

### 2.5. Tissue Preparation and ELISA Measurement of InsP Levels

Frozen tissue was weighed and sonicated in 10 μL/mg of tissue of Tris-HCl buffer (100 mM; pH 7.5) containing 150 mM NaCl, 5 mM EDTA, 1% Triton X-100, and 1% SDS. Homogenates were diluted 1:26 and InsP levels were determined using the IP-One ELISA kit (Cisbio, Codolet, France) according to the manufacturer’s instructions. The average intra- and inter-assay coefficients of variation were 4.68 and 4.83%, respectively.

### 2.6. Immunoblotting 

Western blot analysis of mGlu5 receptors, the α subunit of Gq protein (Gαq), phospholipase-Cβ (PLC-β), and the mGlu5 receptor-interacting protein, norbin was carried out in the dorsal and ventral hippocampus homogenates used for the assessment of PI hydrolysis of 7 mice randomly selected from naive, learner, extinction, and forgetting groups. An aliquot of tissue extract was mixed with a cocktail of protease inhibitors (Sigma-Aldrich, Cat. 2714) and after protein determination, protein lysates (40 μg) were separated by SDS-PAGE electrophoresis and dried on nitrocellulose. The upper part of the membrane was probed with a polyclonal anti-mGlu5 receptor (Abcam, Waltham, MA, USA, Cat. AB76316, dilution 1:5.000), anti-norbin antibody (Abcam, Waltham, MA, USA, Cat. AB88877, dilution 1:700) or anti-phospholipase-Cβ (Santa Cruz, Dallas, TX, USA, Cat. SC5291, dilution 1:1000). The lower part of the membrane was probed with anti-β-actin (Sigma-Aldrich, Cat. A2228, dilution 1:1000) and anti-Gαq (Santa Cruz, USA, Cat. SC136181, dilution 1:500) monoclonal antibodies. Protein expression was detected by the chemiluminescence (ECL) system, visualized with ChemiDoc XRS (Bio-Rad, Hercules, CA, USA) and analyzed by using ImageJ2 2.3.0 software (NIH, Bethesda, MD, USA).

### 2.7. Statistical Analysis

Statistical analyses were conducted with GraphPad Prism software. For the multiple trial experiments (Morris water maze test, acquisition, and extinction), one-way repeated measures ANOVA was conducted to assess the effects of learning or extinction during sessions. One-sample t-test was conducted to assess memory retention in the probe test. A paired t-test was conducted to assess the effect of forgetting. Results of InsP levels were analyzed by two-way ANOVA with the learning paradigms and drug treatments as between factors. Western blot results were analyzed by one-way ANOVA with the learning paradigms as between factors. ANOVA was carried out independently for each brain area (dorsal and ventral hippocampus, prefrontal cortex). Following the ANOVA analysis, Fisher’s LSD *post-hoc* comparisons were performed. Statistical significance was set at *p* < 0.05. All data are expressed as mean ± SEM. 

## 3. Results

### 3.1. Experimental Design and Learning Paradigm in the Morris Water Maze

As expected, animals successfully performed the following acquisition task: the amount of time required to reach the platform significantly decreased across training sessions (day 1–4) in all mice used in the four experiments (Figure 3A–D). On the 5th day, the platform was removed, and we measured the time spent by the animals in the quadrant where the platform had been previously located (probe test). All groups of mice spent more than 15 sec (i.e., more than 25% of the time) in the target quadrant, indicating a preference for this quadrant and long-term memory retention (Figure 3A–D). Mice that spent less than 15 sec in the target quadrant in the probe trial were not used for measurements of mGlu5 receptor-mediated PI hydrolysis and immunoblot analysis.

Two days after the probe trial, the extinction groups (from experiments 3 and 4) were forced to extinguish the acquired memory by performing two non-reinforced trials (without platform) for three consecutive days (days 6–8). In both experiments, about half of the animals successfully extinguished the formerly acquired hidden platform task. As shown in Figure 3E,F, these animals showed a significant reduction in the time spent in the target quadrant (<15 s), indicating a loss of their preference for the target quadrant, which was clearly evident during the probe trial (day 5). 

In experiment 3, the forgetting group was left undisturbed in the cage for about 5 weeks, after which a single probe trial was performed (37 days after the learning session). The time spent by the animals in the target quadrant was evaluated, and Figure 3G shows that only 6 out of 15 mice spent less than 15 s in the target quadrant, indicating loss of memory in only half of the mice. For this reason, we carried out experiment 4 by leaving mice undisturbed in the cages for 7 weeks (i.e., two more weeks with respect to experiment 3). This allowed 15 out of 17 mice (i.e., 88% of mice) to spontaneously lose their preference for the target quadrant (Figure 3H). Only those mice that spent <15 s in the target quadrant after forced or spontaneous memory extinction (from experiments 3 and 4) were used for measurements of mGlu5 receptor-mediated PI hydrolysis and immunoblot analysis.

### 3.2. In Vivo Assessment of mGlu5 Receptor-Mediated PI Hydrolysis 

To examine whether different paradigms of spatial learning might affect mGlu5 receptor signaling, we assessed mGlu5 receptor-mediated PI hydrolysis in the dorsal hippocampus, ventral hippocampus, and prefrontal cortex at the end of the acquisition phase (after the probe test) or at the end of forced or spontaneous extinction of spatial memory. We measured endogenous InsP levels after treating mice with LiCl (100 mg/kg) followed by a challenge with the mGlu5 receptor PAM VU0360172 (30 mg/kg). Treatment with VU0360172 significantly enhanced the InsP formation in the three brain regions (Figure 4A–C), as expected [31]. Data obtained in the four experiments were highly homogenous; therefore, they were subjected to a cumulative analysis. In contrast, VU036072-stimulated PI hydrolysis was blunted in the dorsal hippocampus of learners as compared to all other groups (Figure 4A). Both forced and spontaneous extinction led to a great enhancement of mGlu5 receptor-mediated PI hydrolysis, with VU0360172-stimulated InsP formation being significantly greater with respect to all other groups (naive, controls, and learners), whereas no difference was found between naive and control mice (Figure 4A). In the ventral hippocampus and prefrontal cortex, mGlu5 receptor-mediated PI hydrolysis was significantly reduced in learner mice with respect to all other groups except control mice because, in these two regions, VU0360172-stimulated InsP formation was lower in control than in naive mice (the difference was significant in the prefrontal cortex but not in the ventral hippocampus). As observed in the dorsal hippocampus, VU0360172 stimulated PI hydrolysis to a greater extent in the extinction and forgetting groups in both the ventral hippocampus and prefrontal cortex (Figure 4B,C). 

### 3.3. Immunoblot Analysis of mGlu5 Receptors and Associated Signaling Proteins

In seven mice randomly selected from the naive, learner, extinction, and forgetting groups, we examined protein levels of the mGlu5 receptor, the α subunit of Gq protein (Gαq), phospholipase-Cβ (PLC-β), and the mGlu5 receptor-interacting protein, norbin, in the dorsal and ventral hippocampus and in prefrontal cortex. No significant changes in mGlu5 receptor protein levels were found in learner mice in the three brain regions as compared to naive mice. However, mGlu5 receptor protein levels were significantly enhanced in the ventral hippocampus in the extinction and forgetting groups (only a trend towards an increase was observed in the dorsal hippocampus) (Figure 5A and Figure 6A) and in the prefrontal cortex in the forgetting group (Figure 7A). No significant changes in norbin protein levels were found in the four groups (Figure 5B and Figure 6B) in the dorsal and ventral hippocampus, but an increase was observed in the prefrontal cortex of the forgetting group with respect to the learner and extinction groups (Figure 7B). Interestingly, the PLC-β levels showed an increase in the extinction and forgetting groups as compared to naive mice in the dorsal hippocampus (Figure 5C) and were significantly enhanced in the forgetting group vs. the learner group in the ventral hippocampus (Figure 6C). Finally, the Gαq protein levels were unchanged in the dorsal hippocampus (Figure 5D), whereas in the ventral hippocampus, they were significantly lower in the learner group, as compared to all other groups. Levels were significantly lower in the extinction group as compared to the naive group (Figure 6D). No differences in the PLC-β and Gαq protein levels were found between the four groups in the prefrontal cortex (Figure 7C,D). Uncropped images of blots are shown in Appendix A. The original Western Blot figures can be found at Appendix A.

## 4. Discussion

We were intrigued by the finding that mGlu5 receptor-mediated PI hydrolysis was reduced in control mice with respect to naive mice in the prefrontal cortex (significant reduction) and ventral hippocampus (trend to a reduction), but not in the dorsal hippocampus. Control mice differed from naive mice because they experienced a new, potentially stressful environment, i.e., the water maze. Previous studies have shown that acute exposure to stress causes epigenetic changes in the expression of mGlu5 receptors in the hippocampus (with no distinction between ventral and dorsal hippocampus) [35]. The ventral hippocampus is involved in the response to stress and in the emotional component of learning [9], and the prefrontal cortex is highly vulnerable to stressful conditions [36]. It is possible that a reduced mGlu5 receptor signaling in the prefrontal cortex and ventral hippocampus is instrumental for a better coping to stress in control mice. Prof. Francesco Ferraguti and his associates have found that conditional knock-out of mGlu5 receptors in D1 receptor-expressing neurons in the amygdala, nucleus accumbens, and dorsal striatum enhances coping with escapable and inescapable stress in mice [37]. It will be interesting to examine whether the selective deletion of mGlu5 receptors in the prefrontal cortex and ventral hippocampus affects the response to stress. 

Learner mice showed blunted mGlu5 receptor signaling in all regions examined. Of note, however, the difference between learners and controls was significant only in the dorsal hippocampus (in the other two regions, the reduction of VU0360172-stimulated PI hydrolysis was significant vs. naive but not vs. control mice). These data are divergent from those reported in a previous study in which mGlu5 receptor-mediated PI hydrolysis was enhanced in the hippocampus of rats trained in an eight-arm radial maze [38]. The difference between the two studies may rely on the animal species (mice vs. rats), the method used for the assessment of PI hydrolysis (receptor stimulation in vivo vs. receptor stimulation in brain slices), and the drug used for the stimulation of PI hydrolysis (a selective mGlu5 receptor PAM here vs. ibotenic acid in the previous study). Ibotenic acid behaves as a mixed agonist of mGlu1/2/3/5 and NMDA receptors [39,40], and activation of mGlu1, mGlu3, and mGlu5 receptors might contribute to the overall stimulation of PI hydrolysis produced by ibotenic acid in brain slices [30,41]. 

The blunted mGlu5 receptor-mediated PI hydrolysis found in the dorsal hippocampus of learner mice was not associated with changes in the levels of mGlu5 receptors, the α subunit of Gq proteins, PLC-β, or norbin, a protein that positively modulates mGlu5 receptor signaling [33]. This suggests that learning in the water maze causes a reduction in mGlu5 receptor activity rather than changes in receptor expression or in the expression of signaling proteins, although immunoblot analysis detects the overall levels of mGlu5 receptors in tissue homogenates and not the number of receptors expressed in the plasma membrane. We speculate that mGlu5 receptors are endogenously activated during the learning process and consolidation of memory, and this causes receptor desensitization. Whether changes in receptor signaling are specific to the PI branch or involve other signaling pathways (e.g., the MAP kinase and phosphatidylinositol-3-kinase pathways) remains to be determined. The blunted mGlu5 receptor signaling found in the dorsal hippocampus of learner mice might represent a mechanism of homeostatic synaptic plasticity, which might contribute to learning selectivity and optimization of the signal-to-noise ratio. The evidence that mGlu5 receptors are involved in the mechanisms of synaptic scaling [42,43] supports this hypothesis. 

In the ventral hippocampus and prefrontal cortex of learner mice, mGlu5 receptor-mediated PI hydrolysis was reduced with respect to naive, but not control, mice. This suggests that changes in mGlu5 receptor signaling are caused by exposure to a novel environment rather than learning. This hypothesis is consistent with the evidence that lesions of the ventral hippocampus do not affect spatial learning in the water maze [8]. We also found a reduction in the levels of the α subunit of Gq proteins in the ventral hippocampus of learner mice, suggesting that other Gq-coupled receptors may be affected by learning. 

We found a large up-regulation of mGlu5 receptor-mediated PI hydrolysis in the dorsal and ventral hippocampus of mice with forced or spontaneous extinction of spatial memory. These two conditions reflect different biological processes. Forced extinction induced by six trials in the maze without the platform represents a form of re-learning, in which the association formed during the acquisition phase is suppressed but not erased [44,45]. Different mechanisms underlie spatial acquisition and forced extinction, as shown by the evidence that inhibition of protein synthesis differentially affects the two processes [46]. In contrast, spontaneous extinction likely reflects a progressive weakening of synaptic strength in the circuit engaged during memory acquisition. Thus, we were surprised to find the same changes in mGlu5 receptor signaling in the two paradigms of extinction. The “extinction” and “forgetting” groups of mice showed greater mGlu5 receptor-mediated PI hydrolysis in the dorsal and ventral hippocampus with respect to naive and control mice, suggesting that memory loss was associated with a rebound of mGlu5 receptor function, which exceeded receptor activity in the absence of learning. This phenomenon might be causally related to loss of memory because pharmacological blockade of mGlu5 receptors impairs memory extinction in the water maze [21,22]. Different mechanisms may underlie the up-regulation of mGlu5 receptor-mediated PI hydrolysis in the dorsal and ventral hippocampus of “extinction” and “forgetting” mice. In the dorsal hippocampus, mGlu5 receptor protein levels were unchanged, whereas PLC-β levels were increased in both groups of mice. In contrast, in the ventral hippocampus, mGlu5 receptor expression was up-regulated in both groups, whereas PLC-β levels were exclusively increased in the “forgetting” group. This raises the possibility that memory loss is associated with a functional upregulation of all receptors coupled to PI hydrolysis in the dorsal hippocampus. This hypothesis warrants further investigation. In the prefrontal cortex, we did not find an upregulation of mGlu5 receptor-mediated PI hydrolysis associated with memory loss, although in the “extinction” group, VU0360172-stimulated InsP formation was greater than in control and learner groups. Interestingly, the extinction and forgetting groups differed in the expression of mGlu5 receptors and norbin, which was greater in the forgetting group. Norbin interacts with mGlu5 receptors and plays a permissive role in mGlu5 receptor-mediated PI hydrolysis, calcium oscillations, and MAPK activation [33]. Thus, the increase in mGlu5 receptors and norbin expression in the prefrontal cortex of “forgetting” mice is difficult to reconcile with the lack of changes in mGlu5 receptor-mediated PI hydrolysis. Perhaps, spontaneous memory loss biases mGlu5-mediated signaling towards a PI-independent pathway in the prefrontal cortex. The local infusion of receptor antagonists may help to establish whether the mGlu5 receptors in the prefrontal cortex are involved in the mechanism of memory loss.

## 5. Conclusions 

In conclusion, we have found that spatial memory and memory extinction are associated with robust changes in mGlu5 receptor signaling not only in the dorsal hippocampus (the region encoding the cognitive component of spatial learning), but also in the ventral hippocampus and prefrontal cortex, two regions that are involved in the emotional component of learning. The hypothesis that an up-regulation of mGlu5 receptor signaling is causally related to memory extinction might suggest the use of mGlu5 receptor PAMs to erase pathological memories that translate into psychiatric disorders, as occurs in the pathophysiology of post-traumatic stress disorder. Accordingly, the pharmacological enhancement of mGlu5 receptors has been shown to facilitate contextual fear memory extinction in mice [47].

## Figures and Tables

**Figure 1 life-12-00463-f001:**
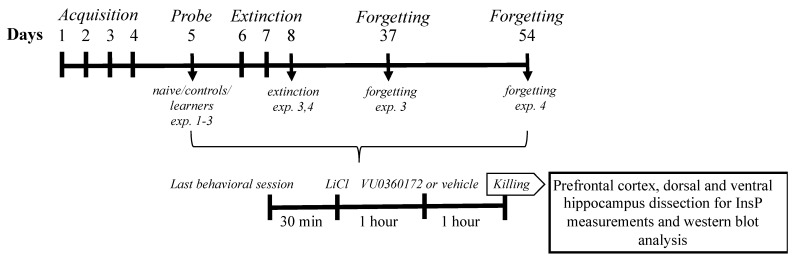
Experimental diagram of the study.

**Figure 2 life-12-00463-f002:**
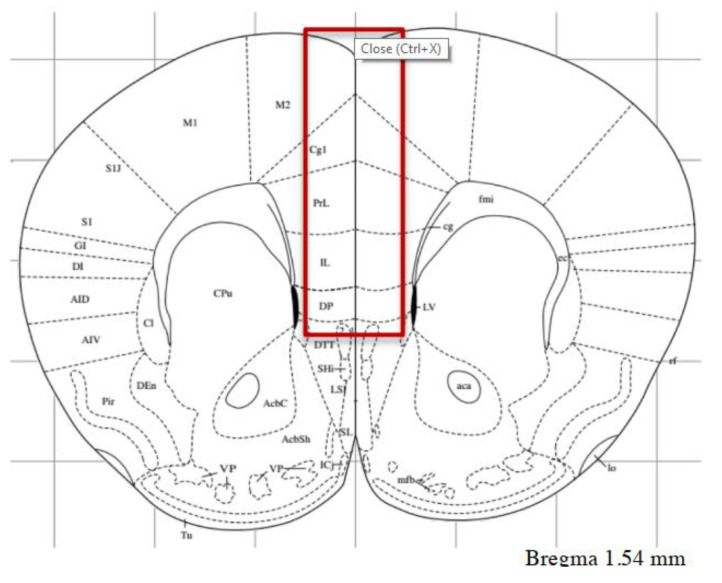
Schematic representation of prefrontal cortex dissection. After removing the olfactory bulbs, the most anterior coronal section of approximately 1 mm was discarded, and the following section of approximately 1 mm (approximately from 2.5 to 1.5 mm anterior to Bregma) was used for dissection of prefrontal cortex (containing a portion of the secondary motor cortex M2). Modified from Paxinos G, Franklin K (2012). The mouse brain in stereotaxic coordinates, Ed 4. San Diego, CA: Academic Press.

**Figure 3 life-12-00463-f003:**
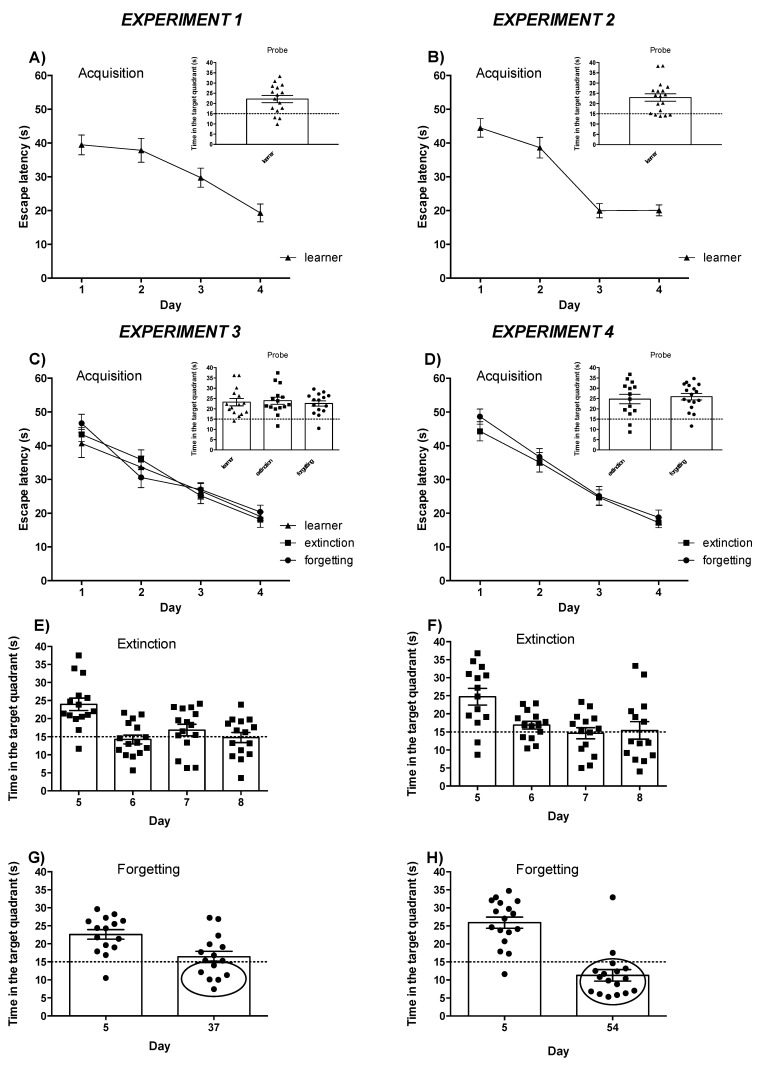
Spatial learning and extinction of memory was assessed in the Morris water maze in four different experiments. (**A**–**D**) Spatial training (acquisition): escape latencies to reach the hidden platform throughout the four days of the task. In the four experiments mice successfully performed the acquisition task indeed the escape latency decreased significantly across training sessions (days 1–4) in all groups of mice tested (learner, extinction and forgetting). One-way repeated measures ANOVA: experiment 1, session effect *F*_(3,299)_ = 10.99, *p* = 0.000001; experiment 2, session effect *F*_(3,259)_ = 42.50, *p* = 0.000001; experiment 3, session effect *F*_(3,899)_ = 71.48, *p* = 0.000001; experiment 4, session effect *F*_(3,619)_ = 61.48, *p* = 0.000001. (**A**–**D**) Probe trial (retention): in the four experiments all groups of mice spent more than 15 s (i.e., more than 25% of time) in the target quadrant indicating a preference for this quadrant and a long-term memory retention. One-sample t-test: each group of mice is significantly different from the threshold of 15 s. Experiment 1: learner, t = 4.08, *p* = 0.001. Experiment 2: learner, t = 4.43, *p* = 0.0004. Experiment 3: learner, t = 4.68, *p* = 0.0004; extinction, t = 5.18, *p* = 0.0001; forgetting, t = 5.73, *p* = 0.0001. Experiment 4: extinction, t = 4.24, *p* = 0.001; forgetting, t = 7.06, *p* = 0.0001. (**E**,**F**) Extinction: the “extinction” groups (from experiments 3 and 4) were forced to extinguish the acquired memory by performing two non-reinforced trials (without platform) for three consecutive days (days 6–8). In both experiments, about half of the animals successfully extinguished the formerly acquired hidden platform task indeed these animals showed a significant reduction of the time spent in the target quadrant (<15 s), indicating a loss of their preference for the target quadrant, clearly evident during the probe trial (day 5). One-way repeated measures ANOVA: experiment 3, day effect *F*_(3,59)_ = 9.82, *p* = 0.0004; experiment 4, day effect *F*_(3,55)_ = 6.58, *p* = 0.009. (**G**,**H**) Forgetting: in the experiment 3 (**G**), a single probe trial was performed 37 days after the acquisition session and only 6 out of 12 mice spent less than 15 sec in the target quadrant (inside the circle), indicating a spontaneous loss of memory only in half of mice. Paired t-test: t = 3.25, *p* = 0.0058. For this reason, we carried out experiment 4 (**H**) in which we tested mice 54 days after the acquisition session (i.e., 2 more weeks with respect to experiment 3). This allowed 15 out of 17 mice (i.e., 88% of mice, inside the circle) to spontaneously lose their preference for the target quadrant. Paired t-test: t = 5.51, *p* < 0.0001.

**Figure 4 life-12-00463-f004:**
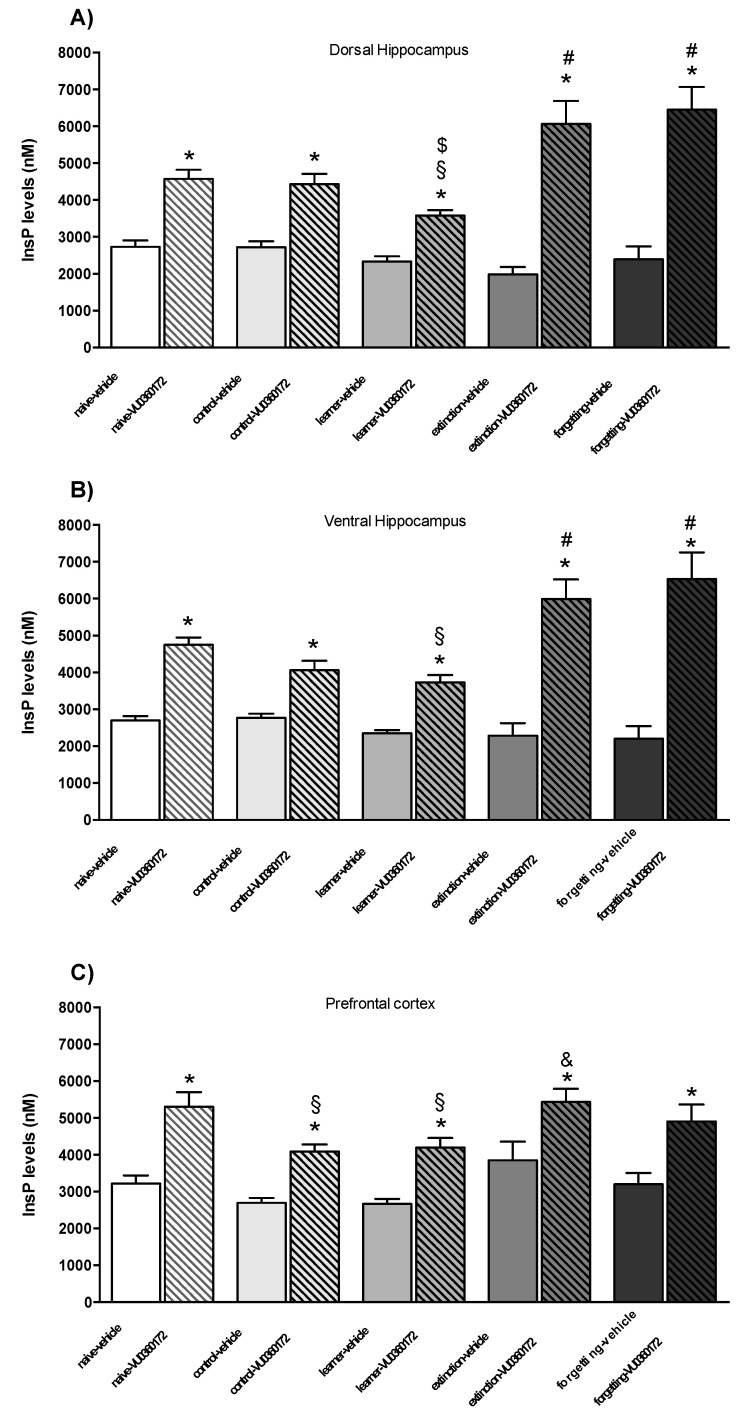
Measurements of endogenous InsP levels in mice treated with lithium ions and then challenged with the selective mGlu5 receptor PAM VU0360172. InsP levels in dorsal hippocampus (**A**), ventral hippocampus (**B**) and prefrontal cortex (**C**) of adult mice treated i.p. with 100 mg/kg of lithium chloride (LiCl) followed by vehicle or VU0360172 (30 mg/kg). Data obtained in the four experiments were highly homogenous and, therefore, were subjected to a cumulative analysis. Values are means ± SEM of 7–24 mice per group. Two-way ANOVA (treatment x learning). Dorsal hippocampus: treatment effect *F*_(1,139)_ = 164.1, *p* < 0.0001; learning effect *F*_(4,139)_ = 6.57, *p* = 0.0002; interaction *F*_(4,139)_ = 8.99, *p* < 0.0001. Ventral hippocampus: treatment effect *F*_(1,139)_ = 137.5, *p* < 0.0001; learning effect *F*_(4,139)_ = 4.75, *p* = 0.0009; interaction *F*_(4,139)_ = 7.30, *p* < 0.0001. Prefrontal cortex: treatment effect *F*_(1,71)_ = 45.26, *p* < 0.0001; learning effect *F*_(4,71)_ = 5.62, *p* = 0.0009. Fisher’s LSD *post-hoc*: * *p* < 0.05 vs. respective vehicle-treated group; # *p* < 0.05 vs. naive-VU0360172, control-VU0360172 and learner-VU0360172 groups; § *p* < 0.05 vs. naive-VU0360172 group; $ *p* < 0.05 vs. control-VU0360172 group; & *p* < 0.05 vs. control-VU0360172 and learner-VU0360172 group.

**Figure 5 life-12-00463-f005:**
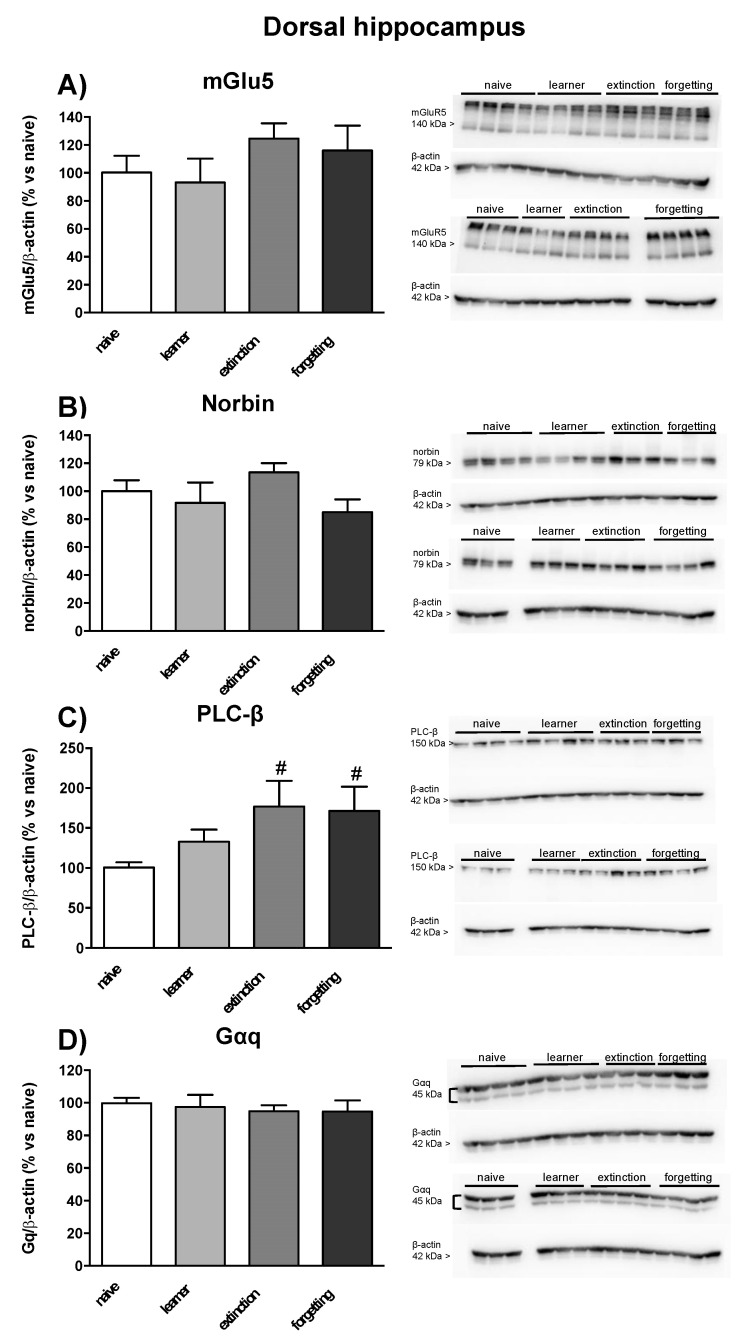
Western blot analysis in dorsal hippocampus of seven mice randomly selected from naive, learner, extinction and forgetting groups used for the assessment of PI hydrolysis. (**A**) mGlu5 receptor, (**B**) mGlu5 receptor-interacting protein, norbin, (**C**) phospholipase-Cβ (PLC-β), and (**D**) α subunit of Gq protein (Gαq) protein levels in the dorsal hippocampus. Each lane depicts the protein expression of a single animal from the four groups. Densitometric analysis (Appendix A) is expressed as percentage of optical density (OD) versus naive group; mGlu5 densitometric analysis was performed on the monomeric band. Individual data points are shown in Appendix A. Densitometric values are means ± SEM. One-way ANOVA + Fisher’s LSD *post-hoc*: learning effect *F*_(3,24)_ = 2.30, # *p* = 0.04 vs. naive group.

**Figure 6 life-12-00463-f006:**
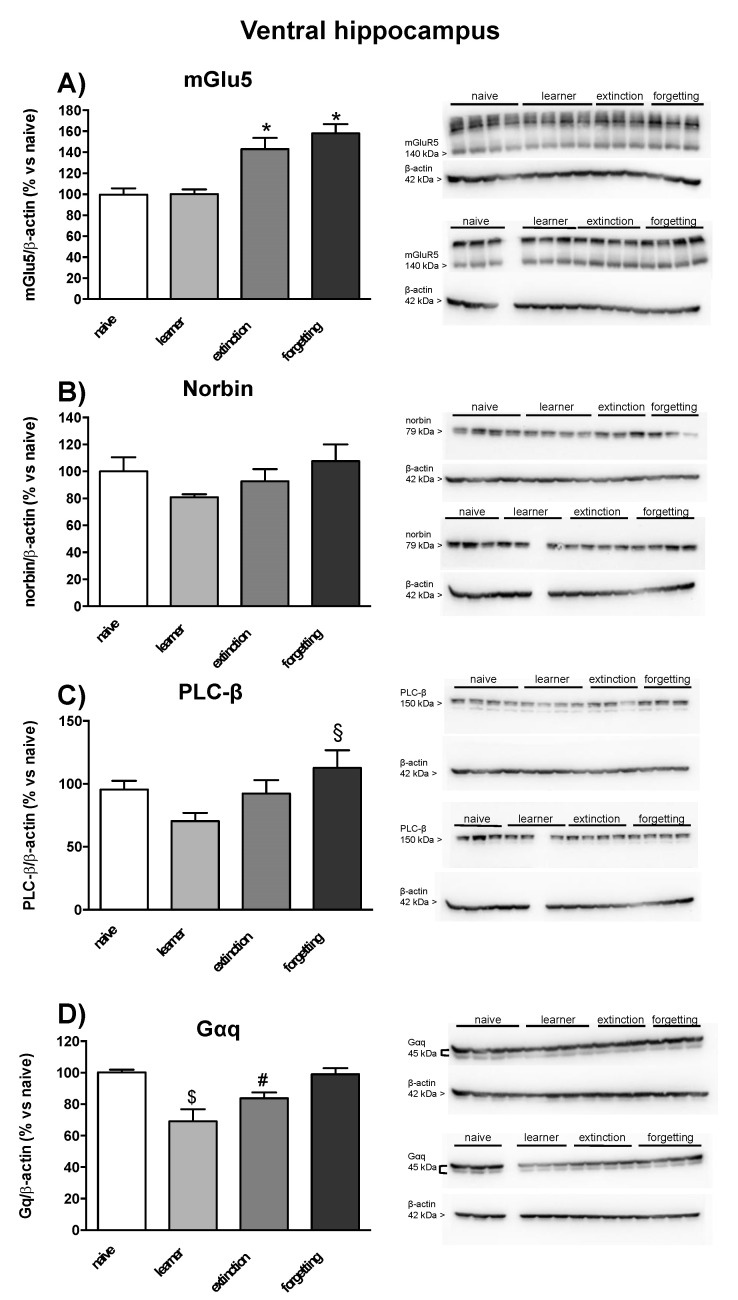
Western blot analysis in ventral hippocampus of seven mice randomly selected from naive, learner, extinction and forgetting groups used for the assessment of PI hydrolysis. (**A**) mGlu5 receptor, (**B**) mGlu5 receptor-interacting protein, norbin, (**C**) phospholipase-Cβ (PLC-β), and (**D**) α subunit of Gq protein (Gαq) protein levels in the ventral hippocampus. Each lane depicts the protein expression of a single animal from the four groups. Densitometric analysis (Appendix A) is expressed as percentage of optical density (OD) versus © group; mGlu5 densitometric analysis was performed on the monomeric band. Individual data points are shown in Appendix A. Densitometric values are means ± SEM. One-way ANOVA + Fisher’s LSD *post-hoc*: mGluR5, learning effect *F*_(3,24)_ = 14.56, * *p* < 0.0001©naive and vs. learner group; PLC-β, learning effect *F*_(3,24)_ = 3.02, § *p* = 0.0063 vs. learner group; Gαq, learning effect *F*_(3,24)_ = 9.71, $ *p* = 0.002 vs. naive, extinction and forgetting groups, # *p* = 0.03 vs. naive and forgetting groups.

**Figure 7 life-12-00463-f007:**
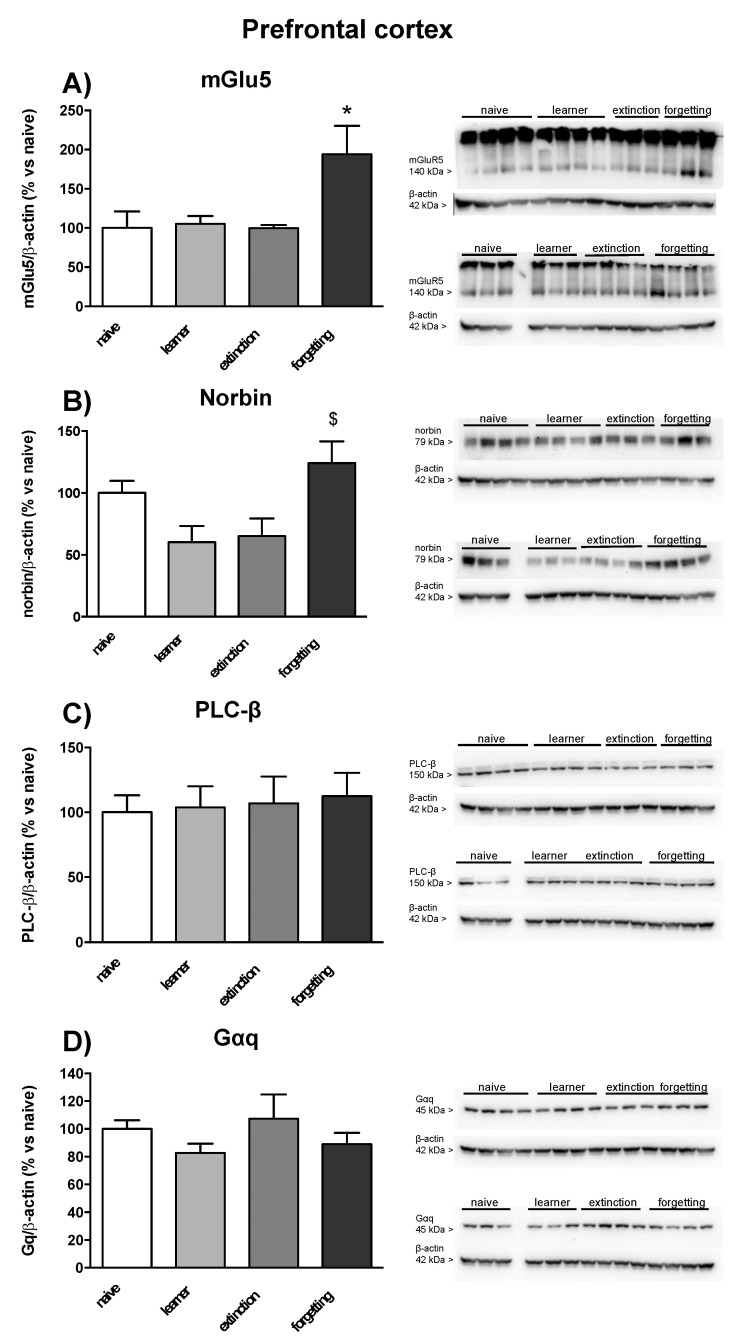
Western blot analysis in prefrontal cortex of 7 mice randomly selected from naive, learner, extinction and forgetting groups used for the assessment of PI hydrolysis. (**A**) mGlu5 receptor, (**B**) mGlu5 receptor-interacting protein, norbin, (**C**) phospholipase-Cβ (PLC-β), and (**D**) α subunit of Gq protein (Gαq) protein levels in the prefrontal cortex. Each lane depicts the protein expression of a single animal from the four groups. Densitometric analysis (Appendix A) is expressed as percentage of optical density (OD) versus naive group; mGlu5 densitometric analysis was performed on the monomeric band. Individual data points are shown in Appendix A. Densitometric values are means ± SEM. One-way ANOVA + Fisher’s LSD *post-hoc*: mGluR5, learning effect *F*_(3,24)_ = 4.58, * *p* = 0.001 vs. all other groups; norbin, learning effect *F*_(3,24)_ = 4.69, $ *p* = 0.01 vs. learner and extinction group.

## Data Availability

The data that support the findings of this study are available from the corresponding author upon reasonable request.

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
