# Peer review of "Changes in mGlu5 Receptor Signaling Are Associated with Associative Learning and Memory Extinction in Mice"

_life, 2022, doi:10.3390/life12030463_

Round 1

Reviewer 1 Report

The topic is interesting and the study is well designed and performed. The data is presented clearly and concise and supportive for their conclusion. However, there are some concerns needed to be addressed before considering to be published:

1) It is better to draw a  experimental diagram to make the experimental process clear;

2) In Fig.3 and Fig.4, statistical column and glue display order is inconsistent. Please revise.

Author Response

The topic is interesting and the study is well designed and performed. The data is presented clearly and concise and supportive for their conclusion. However, there are some concerns needed to be addressed before considering to be published:

1) It is better to draw a  experimental diagram to make the experimental process clear;

As suggested by the reviewer we have now included an experimental diagram as Fig.1

2) In Fig.3 and Fig.4, statistical column and glue display order is inconsistent. Please revise.

We apologize for this, we have now rotated the gels by 180° to make them consistent with the order of the bar graph. 

Reviewer 2 Report

I thoroughly enjoyed reading this article authored by Teleuca et al.  

  1. For Figure 1, might I recommend that they do not label the probe test results (the inserts) as B, D, F, H as (1) it's hard to see those labels in the current figure); (2) the labeling is not intuitive and (2) it's really not necessary.  Instead, I recommend that the authors simply label each panel as A, B, C, D and then when they wish for the reader to look at the inserts, they can use "inserts in Fig.1A-D" as the call-out in the main text.  

2. Given that the Results following immediately from the Methods, it is not necessary to restate the summary of the different experimental groups.  

3. I'm new to reviewing for this journal, so I'm not clear if the statistical results must be placed in the figure legends.  Personally, I would prefer if the statistical results be moved from the figure legends to the main text of the Results, unless this is a violation of the journal format.  Also, please include the p values for all statistical analyses.  F values are insufficient.

4. The authors do not provide statistical results to confirm extinction learning.  Is the time spent in the former platform quadrant significantly lower on later extinction sessions than on Day 1 of extinction or on the probe test conducted following the long periods of no testing?  That is important to determine whether or not the groups are, in fact, behaviorally different.   Similarly, the authors should conduct a Chi-squared analysis to determine if the number of mice exhibiting "forgetting" following 54 days is different from 37 days.  Without knowing how different these groups are from each other, it's difficult to interpret any biochemical findings.

5.  A question I have related to the definition of the "learners":  Is the mean of the mice on the probe test (for those that were trained to find a platform) significantly different from 15 sec (or 25% time)?  This should be determined statistically using one-sample t-tests.  Particularly in Expts 1 and 2, there are quite a few mice hovering around the 15 sec mark, which indicates that some of the mice did not learn (or did not remember what they learned). 

6. I may have missed it in the text, but were all of the animals included in the biochemical assays or only the mice with extreme scores (i.e., only mice exhibiting >15 sec for "learners" and <10 sec for "extinction" or "forgetting"?  I'm curious as to why the sample sizes range from 7 to 24 for the IP3 assay...

7. Was there insufficient tissue to conduct immunoblotting in the PFC?  Also please provide a diagram of the tissue dissection so the reader knows which parts of the PFC were examined.

8. If Figures 3 and 4 could be combined, it would be easier to compare across the brain regions. 

9. line 320, change "receptors" to "receptor"

10. line 323, change "stimulated" to stimulate"

11.  In general, the English of the Discussion is not as strong as the rest of the report and I recommend that the authors carefully proof-read their Discussion prior to submitting the revision.

12. Given that the authors conducted immunoblotting on whole cell homogenates, it should acknowledged that mGlu5 (or any other protein expression) might be changed at the level of the plasma membrane, which could account for their IP3 findings.

13. Given that the PFC was not examined by immunoblotting and particularly the more ventromedial aspect of the PFC has been implicated in extinction learning under operant-conditioning procedures, could it be that the biochemical signature of extinction vs. forgetting lies outside of the hippocampus, likely in the PFC? 

14.  The very long paragraph at the end of the report should be subdivided so the conclusions are highlighted.  

Author Response

I thoroughly enjoyed reading this article authored by Teleuca et al.  

  1. For Figure 1, might I recommend that they do not label the probe test results (the inserts) as B, D, F, H as (1) it's hard to see those labels in the current figure); (2) the labeling is not intuitive and (2) it's really not necessary.  Instead, I recommend that the authors simply label each panel as A, B, C, D and then when they wish for the reader to look at the inserts, they can use "inserts in Fig.1A-D" as the call-out in the main text.

We eliminated labels in the probe test graphs as suggested.   

  1. Given that the Results following immediately from the Methods, it is not necessary to restate the summary of the different experimental groups.  

As suggested by the reviewer we eliminated the summary of experimental groups in the Results.

  1. I'm new to reviewing for this journal, so I'm not clear if the statistical results must be placed in the figure legends.  Personally, I would prefer if the statistical results be moved from the figure legends to the main text of the Results, unless this is a violation of the journal format.  Also, please include the p values for all statistical analyses.  F values are insufficient.

There are not specific indications in the Instructions for Authors, if possible we prefer to leave all statistical analysis in the figure legends to facilitate reading of the Results. As requested we included all p values in the figure legends.

  1. The authors do not provide statistical results to confirm extinction learning.  Is the time spent in the former platform quadrant significantly lower on later extinction sessions than on Day 1 of extinction or on the probe test conducted following the long periods of no testing?  That is important to determine whether or not the groups are, in fact, behaviorally different.   Similarly, the authors should conduct a Chi-squared analysis to determine if the number of mice exhibiting "forgetting" following 54 days is different from 37 days.  Without knowing how different these groups are from each other, it's difficult to interpret any biochemical findings.

We now added statistical analysis to confirm extinction learning and forgetting. Forgetting data of 37 and 54 days originate from two different experiments. In the 37-day experiment, statistical analysis with the paired t test gave a p value of 0,0058, whereas in the 54-day experiment the p value was < 0,0001 as compared to values obtained at the end of learning (day 5). Thus, statistical analysis indicates that 54 days represent an optimal forgetting time. For the assessment of PI hydrolysis we used exclusively those animals that were below the empirical threshold of 15 sec from both experiments. All this is now included in the legend of Fig. 3 (previously named Fig 1).

  1. A question I have related to the definition of the "learners":  Is the mean of the mice on the probe test (for those that were trained to find a platform) significantly different from 15 sec (or 25% time)?  This should be determined statistically using one-sample t-tests.  Particularly in Expts 1 and 2, there are quite a few mice hovering around the 15 sec mark, which indicates that some of the mice did not learn (or did not remember what they learned). 

Performing one-sample t-test on probe test we obtain that each group of mice was significantly different from 15 sec. Exp 1: learner, t = 4.08, p = 0.001. Exp 2: learner, t = 4.43, p = 0.0004. Exp 3: learner, t = 4.68, p = 0.0004; extinction, t = 5.18, p = 0.0001; forgetting, t = 5.73, p = 0.0001. Exp 4: extinction, t = 4.24, p = 0.001; forgetting, t = 7.06, p = 0.0001. We add these statistical results also in the manuscript (Legend of Figure 3).

  1. I may have missed it in the text, but were all of the animals included in the biochemical assays or only the mice with extreme scores (i.e., only mice exhibiting >15 sec for "learners" and <10 sec for "extinction" or "forgetting"?  I'm curious as to why the sample sizes range from 7 to 24 for the IP3 assay…

As we wrote in the “Results” , paragraph 3.1, only those mice that spent <15 sec in the target quadrant after forced or spontaneous memory extinction (from experiments 3 and 4) were used for measurements of mGlu5 receptor-mediated PI hydrolysis and immunoblot analysis. We missed to write that, for what concern the learner groups, only mice that spent >15 sec in the probe trial were included in the biochemical measurements. We apologize for this and we have now included this sentence in the manuscript (paragraph 3.1).

The sample size ranges from 7 to 24 because of the selection of the mice with the correct scores, i.e. in the exp 3 only 7 mice are under the threshold of 15 sec (Fig.3E), in the exp 4 only 8 mice (Fig 3F) for a total of 15 mice, half of which were treated with vehicle and half with VU0360172 for the measurements of InsP.

  1. Was there insufficient tissue to conduct immunoblotting in the PFC?  Also please provide a diagram of the tissue dissection so the reader knows which parts of the PFC were examined.

We have now included a diagram of tissue dissection (with the limits inherent to hand dissection) in the new figure 2. We had no time to perform Western analysis in the PFC to respect the submission deadline in the first version of the manuscript. Having enough residual tissue we have now perform this analysis and new data are shown in figure 7, presented in the results section, paragraph 3.3, and discussed at page 15.

  1. If Figures 3 and 4 could be combined, it would be easier to compare across the brain regions. 

Thank you for the suggestion but now we have added also figure 7 for PFC immunoblotting so combining all the blots from the 3 brain regions in a single figure would result in a too crowded figure.

  1. line 320, change "receptors" to "receptor"

Thank you. It is done.

  1. line 323, change "stimulated" to stimulate"

Thank you. It is done.

  1. In general, the English of the Discussion is not as strong as the rest of the report and I recommend that the authors carefully proof-read their Discussion prior to submitting the revision.

We are sorry for that. The discussion has been extensively revised.

  1. Given that the authors conducted immunoblotting on whole cell homogenates, it should acknowledged that mGlu5 (or any other protein expression) might be changed at the level of the plasma membrane, which could account for their IP3 findings.

Yes indeed. This is now highlighted in the discussion section, page 14.

  1. Given that the PFC was not examined by immunoblotting and particularly the more ventromedial aspect of the PFC has been implicated in extinction learning under operant-conditioning procedures, could it be that the biochemical signature of extinction vs. forgetting lies outside of the hippocampus, likely in the PFC? 

This was now examined and discussed.

  1. The very long paragraph at the end of the report should be subdivided so the conclusions are highlighted. 

Thank you. It is done.